# Genome-Wide Identification of the *Paulownia fortunei* Aux/IAA Gene Family and Its Response to Witches’ Broom Caused by Phytoplasma

**DOI:** 10.3390/ijms25042260

**Published:** 2024-02-13

**Authors:** Jiaming Fan, Minjie Deng, Bingbing Li, Guoqiang Fan

**Affiliations:** 1College of Forestry, Henan Agricultural University, Zhengzhou 450002, China; fanjiaming1114@163.com (J.F.); dengmj1980@henau.edu.cn (M.D.); libing20160727@163.com (B.L.); 2Institute of Paulownia, Henan Agricultural University, Zhengzhou 450002, China

**Keywords:** *Paulownia fortunei*, auxin, protein–protein interaction, Paulownia witches’ broom, salicylic acid

## Abstract

The typical symptom of Paulownia witches’ broom (PaWB), caused by phytoplasma infection, is excessive branching, which is mainly triggered by auxin metabolism disorder. *Aux/IAA* is the early auxin-responsive gene that participates in regulating plant morphogenesis such as apical dominance, stem elongation, lateral branch development, and lateral root formation. However, no studies have investigated the response of the *Aux/IAA* gene family to phytoplasma infection in *Paulownia fortunei*. In this study, a total of 62 *Aux/IAA* genes were found in the genome. Phylogenetic analysis showed that *PfAux/IAA* genes could be divided into eight subgroups, which were formed by tandem duplication and fragment replication. Most of them had a simple gene structure, and several members lacked one or two conserved domains. By combining the expression of *PfAux/IAA* genes under phytoplasma stress and SA-treated phytoplasma-infected seedlings, we found that *PfAux/IAA13/33/45* may play a vital role in the occurrence of PaWB. Functional analysis based on homologous relationships showed a strong correlation between *PfAux/IAA45* and branching. Protein–protein interaction prediction showed that PfARF might be the binding partner of PfAux/IAA, and the yeast two-hybrid assay and bimolecular fluorescent complementary assay confirmed the interaction of PfAux/IAA45 and PfARF13. This study provides a theoretical basis for further understanding the function of the *PfAux/IAA* gene family and exploring the regulatory mechanism of branching symptoms caused by PaWB.

## 1. Introduction

Auxin, a widely distributed plant hormone, is a key regulator of plant growth and development processes, including embryogenesis, branch elongation, directional response, and flower and fruit development [1,2,3,4]. The early auxin-responsive genes include three gene families: *SAUR*, *Aux/IAA*, and *GH3* [5]. Among them, the *Aux/IAA* gene family plays an important role in auxin signaling and in regulating the expression of downstream genes [6]. *Aux/IAA* encodes nuclear proteins with short half-lives that typically have four signature conserved domains: Domain I, II, III, and IV [7,8]. Domain I is rich in the conserved leucine motif LxLxL, which is necessary for Aux/IAA proteins to perform transcriptional inhibitory functions [9]. Domain II is associated with protein stability and mediates ubiquitination degradation of Aux/IAA by recognizing F-box protein TIR1 [10]. Domain III and IV are responsible for the formation of homologous dimerization between Aux/IAA proteins and heterodimerization with ARF (auxin response factor) [5]. Auxin perception and signal transduction in plant cells are mainly realized through the TIR1/AFB-Aux/IAA-ARF pathway [11]. When auxin concentration is low, a heterodimer will be formed between Aux/IAA and ARF proteins, resulting in transcription suppression of ARF-targeted genes [12,13]. When auxin concentration is high, the SCF-TIR1 complex targets Aux/IAA protein for ubiquitination degradation, whereby the ARF proteins are released and their target genes are activated [14,15]. Different Aux/IAA dimer binding activities vary greatly, which allows the Aux/IAA proteins to respond to different auxin concentrations [16].

Several *Aux/IAA* gene families have been reported as playing important roles in plant growth and development and responding to abiotic and biotic stress. *SlIAA9* was involved in leaf morphological changes and parthenocarpy in tomato [17]. The *MeAux/IAAs*-silenced plants showed disease sensitivity against Xam in cassava [18]. In orchard grass, *DgIAA21* was involved in drought stress [19]. The apple callus overexpressing *MdIAA8/9/25* was tolerant to salt stress [20]. In peach, *PpIAAs* participated in governing texture and hardness in stone fruit maturation [21]. In addition, auxin is a key component of plant signaling in response to biotic stress. In Arabidopsis, the auxin-stimulated SCF ubiquitination pathway has been shown to be directly related to the resistance of necrotrophic pathogens [22], and to regulate the expression of bacterial virulence genes, increasing plant susceptibility to PtoDC3000 pathogenesis [23]. The P2 protein hijack of Rice Dwarf Virus (RDV) can bind OsIAA10 protein, thereby inhibiting 26S proteasome degradation of OsIAA10 and making rice more susceptible to RDV infection [24].

*Paulownia fortunei*, native to China, is a multipurpose and fast-growing tree species, which is widely planted in temperate zones [25]. However, it is susceptible to infection by the phytopathogen phytoplasma, bringing about the occurrence of Paulownia witches’ broom (PaWB). The infected plants show witches’ broom and dwarfism, and PaWB is highly infectious, which seriously affects wood production efficiency [26]. The occurrence of PaWB is caused by a reduction in auxin content, and after IAA treatment, the branching number is reduced, and at the same time, the key genes in auxin signal transduction also change [27]. A systematic study of the *PfARF* gene family found that some genes were significantly expressed after PaWB formation [28]. Taking account of the cooperation of Aux/IAA and ARF proteins in auxin signaling, we presumed *PfAux/IAA* gene family members might respond to PaWB. To date, a systematic study of the *P. fortunei Aux/IAA* gene family has not yet been reported. Therefore, the aim of this study was to analyze the molecular characteristics of *PfAux/IAA* genes and their evolutionary relationships, and then identify the correlation between *PfAux/IAAs* and PaWB. We used bioinformatics methods to identify the *PfAux/IAA* gene family based on the genome information *(Paulowina fortunei Genome V1.0)*, analyzed the molecular characteristics and transcript level of the *PfAux/IAA* genes during the occurrence of PaWB, verified the interaction of several Aux/IAA and ARF proteins, and explore the response of the *PfAux/IAA* genes to phytoplasma. These results will provide a basis for further research on the function of the *PfAux/IAA* genes, and give clues for an in-depth understanding of the pathogenesis of PaWB. 

## 2. Results

### 2.1. Identification of the PfAux/IAA Gene Family and Analysis of Protein Properties

In this study, a total of 62 *PfAux/IAA* gene family members were identified (Appendix A**)**, named *PfAux/IAA1-PfAux/IAA62* according to their position on the chromosome (Figure 1). There were 60 *PfAux/IAA* genes unevenly distributed on 19 chromosomes of *P. fortunei*, with the majority distributed on chromosome 08, and most of the *PfAux/IAA* genes distributed at both ends of the chromosome.

The analysis of properties of PfAux/IAA proteins showed (Appendix A) that PfAux/IAA had great differences in length, and the shortest one was 179 amino acids (PfAux/IAA48), the longest one was 1240 amino acids (PfAux/IAA1). The molecular weight of the protein ranged from 40.67 to 270.48 kD, and the theoretical isoelectric point ranged from 5 to 9.24. Except for four members (PfAux/IAA4, PfAux/IAA33, PfAux/IAA59, PfAux/IAA35), which were stable proteins (instability index < 40), the rest of the family members were unstable hydrophilic proteins (instability index > 40, gravy < 0). In addition, 54 members were predicted to be located in the nucleus, 2 in the vacuole, and 6 in the chloroplast, suggesting that members of this family might play roles in different cellular environments, and identifying their associated molecular properties would help in the study of their biological functions.

### 2.2. Phylogenetic Tree Construction and Collinearity of PfAux/IAA Genes

The phylogenetic tree was constructed based on the alignment analysis of 29 AtAux/IAA and 62 PfAux/IAA proteins. The results showed that the *Aux/IAA* gene family in *P. fortunei* was grouped in a similar way to *A. thaliana,* which could be divided into A and B groups (Figure 2). Each group was further divided into four subgroups, among which the B4 subgroup contained the most members and the A4 subgroup contained the least. The similarity in the classification of subgroups suggested a certain evolutionary conservatism in the *Aux/IAA* family. Compared with those of *A. thaliana*, the B1 and B4 subgroups were significantly expanded. In particular, the B4 subgroup had only 3 members in *A. thaliana*, but 30 members in *P. fortunei*.

To further understand the evolutionary relationship of *PfAux/IAA* genes, we analyzed duplication events occurring in these genes (Figure 3A). *PfAux/IAA13* and *PfAux/IAA 14*, *PfAux/IAA43* and *44*, *PfAux/IAA48* and *49*, *PfAux/IAA22/23/24* and *25*, *PfAux/IAA33* and *34*, *PfAux/IAA37* and *38,* and *PfAux/IAA57* and *58*, respectively, formed seven tandem replication regions inside chromosomes. The 50 *PfAux/IAA* genes, distributed in the duplicated regions of the genome at both ends of the chromosome, participated in 44 fragmental replication events, 13 of which occurred among tandemly duplicated genes. The “copies” of tandem duplication genes in the subgroup A1 caused the formation of the A3 subgroup. These results suggested that the overall expansion of *PfAux/IAA* genes might be caused by the combination of fragment replication and tandem replication events, and was involved in genome-scale replication events.

Collinearity analysis of *P. fortunei* and the *A. thaliana Aux/IAA* gene family indicated (Figure 3B) that fragment duplication events existed between 48 *PfAux/IAA* genes and 29 *AtIAA* genes. Each of the eleven *PfAux/IAA* genes simultaneously shared homologous relationships with segments of more than three *AtIAA* genes*,* which meant the *Aux/IAA* family genes might undergo multiple duplication events and complex gene rearrangements during the evolution of plants from herbaceous to woody plants.

### 2.3. Gene Structure and Motif Analysis of PfAux/IAA Proteins

Conserved motif analysis of Aux/IAA proteins using the online tool MEME (Appendix A) indicated that motifs 16, 11, 4, and 9 corresponded to four conserved domains of canonical Aux/IAA proteins: Domain I, II, III, and IV respectively. The 26 members of group A as well as subgroups B1 and B2 all had four conserved domains. Most of the subgroups B3 and B4 were missing one or two conserved domains. Except for PfAux/IAA8, 61 remaining members contained Domain IV (motif9). Furthermore, motif2 and 12 were present in most of the PfAux/IAA proteins, and the nine remaining motifs were mostly present in the members of the subgroup B4. 

The schematic gene structures of *PfAux/IAA* genes using the GSDS server displayed that the length of coding and non-coding regions of these members varied greatly; a few (11) genes did not contain non-coding regions (untranslated region, UTR) (Appendix A). The structure of most genes was relatively simple, less than 4 kb in length; the number of exons ranged from two to five, and the number of introns varied from two to four. Most members of the B4 subgroup were more than 5 kb in length and had a high number of introns and exons. The numbers of exons/introns of the same subgroup were not similar.

### 2.4. Promoter Region cis-Element Prediction of PfAux/IAA Genes

Analysis of the cis-elements of gene promoters can help us understand gene expression patterns and predict possible biological functions. We extracted the upstream 2000 bp sequences of the start codons of *PfAux/IAA* genes as the promoter regions, and used the PLANTCARE database for cis-acting element analysis. The cis-acting elements of *PfAux/IAA* genes can be categorized into five main groups based on their biological functions: light-responsive elements (462, 42%), growth and development elements (54, 5%), hormone-responsive elements (350, 31%), metabolic responsive elements (156, 14%), and stress-responsive elements (90, 8%) (Figure 4A). We discovered that all *PfAux/IAA* genes except *PfAux/IAA48* and *PfAux/IAA31* contained cis-elements associated with hormone-responsive, including jasmonic acid, gibberellin, abscisic acid, auxin, and salicylic acid, which is in line with the report that the *Aux/IAA* gene family is involved in the synergistic regulation of phytohormone signaling (Figure 4B) [29]. In addition, 48 *PfAux/IAA* genes contained stress-responsive elements, suggesting that they may have specific stress resistance.

### 2.5. Expression Analysis of PfAux/IAA Genes in Answer to Phytoplasma

Our preliminary study found that the PaWB gradually returned to normal phenotype after treatment with methyl methane sulfonate (MMS) at the appropriate concentration (Appendix A). In this study, we aimed to find some *PfAux/IAA* genes that might respond to the infection of phytoplasma according to transcriptome data of PaWB seedlings treated with MMS (Figure 5A). A total of 49 of 62 *PfAux/IAA* genes were detected in healthy (PF) and diseased seedlings (PFI). Eight genes *PfAux/IAA13/14/33/34/37/38/45/58* were highly expressed (fold change > 1.5) in PFI, which may be a positive response to the regulation of phytoplasma, and in one gene *PfAux/IAA29,* the expression was differentially reduced (fold change > 1.5), which may be a negative response to the regulation of phytoplasma. It is noteworthy that the extension of the MMS (20 mg L^−1^) treatment duration resulted in a down-regulation of the expression levels of the six genes (*PfAux/IAA13/33/37/38/45/58*) that had a positive response to phytoplasma, reaching a level comparable to that observed in healthy seedlings. The outcomes suggested these *PfAux/IAA* genes may be closely associated with the occurrence of PaWB. Expression of these six *PfAux/IAAs* in different tissues indicated that they are constitutive genes, which are expressed in all tissues. As shown in Figure 6A, all *PfAux/IAA* genes had low expression in roots. In particular, *PfAux/IAA45* and *PfAux/IAA58* were highly expressed in stems. To verify the data of RNA-seq, the expression of six *PfAux/IAA* genes was tested by qRT-PCR using healthy and diseased seedlings. As shown in Figure 5B, the transcriptome data were correlated with qRT-PCR results, and their expression trend was similar.

Much experimental evidence has shown that the SA and IAA signal transduction pathways seem to cross and interact [30]. To further explore the expression pattern of differentially expressed *PfAux/IAA* genes following phytoplasma infection, we examined the relative expression of *PfAux/IAA* genes in SA-treated PaWB-infected seedlings and untreated PaWB-infected control seedlings (Appendix A). The results showed (Figure 6B) that the expression of *PfAux/IAAs* exhibited different changes under the SA treatment. Notably, the expression of *PfAux/IAA13/33/45* was reduced and similar to that in healthy seedlings. These results suggested that *PfAux/IAA13/33/45* may act as defense genes in response to SA and play an important role in the joint action of SA and IAA. 

### 2.6. Interaction Analysis of PfAux/IAA 

To gain further insights into the function of *PfAux/IAA13/33/45*, a protein interaction network was predicted (Figure 7A). In Arabidopsis, phytoplasmal effector TENGU is known as an inducer of witches’ broom and dwarfism [31], and it represses the expression of the *ARF6* gene [32]. According to study [27], it has been demonstrated that transcription factors *ARFs* play vital roles in PaWB pathogenesis as target genes of microRNA156. As shown in Figure 7A, ARFs have a strong relationship with Aux/IAAs. Since *PfARF13* is the homolog of *AtARF6*, there is considerable occurrence of witches’ broom and dwarfism induced by TENGU. Thus, we conducted the protein–protein interaction to investigate its relationship with PfAux/IAAs. The yeast two-hybrid experiments showed that the transform cells with PfAux/IAA45 and PfARF13 could be grown on SD/-Trp/-Leu and SD/-Trp/-Leu/-Ade/-His+X-a-gal (Figure 7B), which demonstrated that there was a strong physical interaction between them. To further verify the interaction, we then performed the bimolecular fluorescent complementary assay in *N. benthamiana* leaves; the result showed that the co-expression of PfAux/IAA45 with PfARF13 resulted in a strong YFP fluorescence signal in both the nucleus and cell membranes, while no YFP fluorescence signal was observed in the controls (Figure 8). 

## 3. Discussion

Previous studies have shown that expansion of the *Aux/IAA* gene family is associated with whole genome duplication events in plants [33,34,35]. Soybean underwent two genome-wide duplication events and one genome-wide triplication event, and more than 75% of its *Aux/IAA* genes were generated by gene duplication events [33,34]. After two whole-genome duplication events in wheat, the *Aux/IAA* family members were expanded from 28 to 84 [35]. The genome of *P. fortunei.* underwent two genome replication events during its evolution: a genome-wide replication event and a diploidy event [36]. The *PfAux/IAA* family obtained in this study had 62 members, most of which were located in the genome replication regions. Genome-wide replication events have occurred in all four plants, which might account for the expansion of the *Aux/IAA* gene family. 

Typical Aux/IAA proteins have four conserved domains (I–IV), but some members lack one or more of these conserved domains and are called atypical members [9,37]. Protoplast transfection experiments demonstrate that domain I is required for recognizing *ARF* factors as a repressor [9]. So, we speculated that 15 PfAux/IAA proteins lacking domain I might not have a repressive effect on downstream genes. The remaining members having domains III and IV would play roles in binding other Aux/IAA or ARF proteins to form homologous or heterodimers, which was confirmed by the interaction experiments in this study. Meanwhile, Arabidopsis IAA32/34 and IAA33 lacking domain II cannot be recognized by TIR1/AFB, regulating auxin signaling by TMK1-IAA32/34-ARFs and MPK14-IAA33-ARFs pathways, respectively [38,39]. In this case, we presumed 25 PfAux/IAA proteins lacking domain II would participate in auxin-regulated biological processes through other pathways. 

Generally, tissue expression specificity is closely related to function*. GmIAA27* had a higher transcript abundance in stems compared with other vegetative organs and participated in dwarfing and multiple-branching in soybean [4]. *SlIAA17*, which, controlling for fruit size and the thickness of the pericarp, showed 10-fold transcripts in developing fruit compared with the level in flowers [40]. Analogously, the transcript level of *StIAA2,* which is involved in petiole hyponasty, was high in the petiole [41]. Here, we found six *PfAux/IAA* genes (*PfAux/IAA13/33/3738/45/58*) that may be related to the occurrence of PaWB disease. Of them, *PfAux/IAA33* expressed highly in leaves, and *PfAux/IAA45* and *PfAux/IAA58* had the same expression in stems, inferring that they may play a regulatory role in the morphogenesis of leaves and stems.

Previous studies have shown that auxin and SA signaling pathways are intertwined [42,43]. In Arabidopsis, *P. syringae* strain DC3000 produced IAA to suppress the SA-mediated defense [44]. In cotton, the miR393 module regulated plant resistance by auxin perception, and the signaling was SA-dependent [45]. *GhIAA43*-silenced cotton seedlings showed enhanced wilt resistance, accompanied by increased SA contents and activation of maker genes of the SA pathway [46]. Since *Aux/IAAs* are essential in controlling the expression of the gene responsive to auxin, we were interested in determining the response of *Aux/IAAs* to the SA-related defense of PaWB. Similar to the case of *GhIAA43*-silenced cotton plants, our experiments showed that the expression levels of *PfAuxIAA13*, *PfAuxIAA33,* and *PfAuxIAA45* reduced in PaWB-diseased seedlings after treatment with SA. At the same time, growth inhibition resulting from the overaccumulation of defense signals was observed [47]. Moreover, SA-responsive elements were found in the promoter regions of three of the *PfAuxIAA* genes mentioned above; therefore, we hypothesize that several *PfAux/IAA* genes are involved in the SA-induced defense responses.

Experimental evidence showed that heterodimerization between Aux/IAA and ARF proteins has a special function. Zhang, et al. (2021) found that *AtARF4* and *AtARF5* jointly regulated bud regeneration by combating the inhibition of At*IAA12* [48]. Auxin signaling mediated by *AtIAA14* and *AtARF7/19* has important effects on the expression of genes regulating membrane lipid remodeling under low phosphorus stress [49]. In Arabidopsis, the overexpression of TENGU, a pathogenic effector protein homologous to PaWB effector (orf00344-281), decreased the expression level of *AtARF6* [32,37]. *AtIAA8* regulates lateral branches and floral organ development in Arabidopsis [50]. In our study, AtARF6 and PfARF13 belonged to the same branch in the phylogenetic tree [28], and AtIAA8 had a close evolutionary relationship with PfAux/IAA45 (Figure 2). We confirmed the interaction between PfARF13 and PfAux/IAA45 via Y2H and BiFC assays, which were predicted in the protein–protein interaction network. Moreover, *PfAux/IAA45* expression was significantly up-regulated and *PfARF13* expression was significantly down-regulated in PaWB-diseased seedlings [28], which is consistent with the expression trend of *SlARFs* and *SlIAA9* in tomato [51]. Therefore, we speculated that *PfAux/IAA45* regulated *PfARF13* in the normal way, resulting in the symptom of witches’ broom.

In conclusion, a total of 62 family members of *PfAux/IAA* were identified in the *P. fortunei* genome, and the subcellular localization of most members was predicted to be in the nucleus. The transcriptome and qRT-PCR results showed that six *PfAux/IAA* genes (*PfAux/IAA13/33/37/38/45/58)* were related to the occurrence of PaWB disease; among them, three *PfAux/IAAs: PfAux/IAA13*, *PfAux/IAA33,* and *PfAux/IAA45,* were down-regulated under SA-treated PaWB-diseased seedlings involved in SA signaling in response to phytoplasma. Based on the integration of our findings and previous reports, it was hypothesized that the function of PfARF13 was inhibited through the formation of the PfARF13-PfAux/IAA45 complex, leading to the dwarf and more-branched phenotype of PaWB-diseased Paulownia. Overall, these discoveries contributed to the understanding of the *Aux/IAA* gene family in *Paulownia fortunei.* and may provide new insights into explaining the occurrence and phenotype of PaWB.

## 4. Materials and Methods

### 4.1. Planting Materials and Treatments

The planting materials used in this study, including healthy and PaWB-diseased Paulownia seedlings, were all from Paulownia Tree Biotechnology Laboratory of Henan Agricultural University. The plantlets grown for 30 days in 1/2 medium were cut and the terminal buds were transferred to a culture flask containing 100 ml of 1/2 medium (with or without appropriate reagents) and cultured for 30 days. The cultured conditions were 25±2 °C and the light conditions were 130 μmol-m-2 s-1 with a 16/8 h light/dark photoperiod.

Five samples were used to analyze the *PfAx/IAA* gene expression change, including healthy *P. fortunei* seedlings (PF), diseased seedlings infected by phytoplasmas (PFI), and diseased seedlings treated with 20 mg·L^−1^ methyl methane sulfonate (MMS) for 5 days (PFIM20-5), 10 days (PFIM20-10), and 30 days (PFIM20-30), respectively. 

SA Treatment: PaWB-diseased seedlings were treated by adding 13.8 mg·L^−1^ salicylic acid to the medium for 30 days. 

### 4.2. Identification of the P. fortunei Aux/IAA Gene Family

Two methods were used to identify the *PfAux/IAA* gene family. Method A: The Aux/IAA protein sequences of *Arabidopsis thaliana* were downloaded from the NCBI database (https://www.ncbi.nlm.nih.gov/ (accessed on 14 May 2023)), and used as the query sequences for BLASTP searches in the *P. fortunei* genome. Homologous proteins of the *PfAux/IAA* gene family were obtained and the redundancies were removed. Method B: We downloaded hidden Markov model (HMM) of Aux/IAA domain (Pf02309) from Pfam database (http://Pfam.xfam.org/ (accessed on 14 May 2023)), and used HMMER3.0 software to retrieve sequences with this model in the *P. fortunei* protein database (E value 10^5^). Finally, we selected the shared sequences obtained by the above two methods as members of the *PfAux/IAA* gene family. 

These genes were mapped to different chromosomes by TBtools (version 1.0120) [52] and named according to their chromosome position. The online tools ExPASy-ProtParam (https://web.expasy.org/protparam/ (accessed on 23 May 2023)) and WoLFPSORT (https://www.genscript.com/wolf-psort.html (accessed on 23 May 2023)) were used to predict the basic characteristics and the subcellular location of the PfAxu/IAA proteins, respectively. The instability index was used to assess protein stability, a value of less than forty indicates a stable protein and a value of more than forty indicates an unstable protein. Gravity was used to judge the hydrophilicity of the protein, where less than 0 is a hydrophilic protein, and more than 0 is a non-hydrophilic protein.

### 4.3. Phylogenetic Tree Construction and Collinearity of PfAux/IAA Genes

A Maximum Likelihood phylogenetic tree was constructed with MEGA (version 7. 0) [53]. ClustaW was used in multiple sequence alignments. MCScanX was used in analyzing tandem duplication and fragment replication events. The collinearity was visualized by TBtools (version 1.0120).

### 4.4. Gene Structure and Motif Analysis of PfAux/IAA Genes

Motif prediction was performed by MEME online tool (https://memesuite.org/meme/tools/meme (accessed on 18 Junly 2023)), the number of motifs and the maximum number of amino acids were respectively set to 16 and 102, and the rest parameters were default. A gene feature visualization server (GSDS) (http://gsds.cbi.pku.edu.cn/ (accessed on 18 Junly 2023)) was employed for coding region analysis. 

### 4.5. Cis-Elements Prediction of PfAux/IAA Genes

The putative promoter sequences of *PfAux/IAA* genes (2000-bp sequence upstream of the start codon) were extracted with TBtools; PlantCARE (http://bioinformatics.psb.ugent.be/webtools/plantcare/html/ (accessed on 3 January 2024)) was used to search cis-acting elements in these promoter sequences.

### 4.6. Analysis and Validation of PfAux/IAA Genes Expression

We downloaded the RNA-seq data of PF, PFI, PFI20-5, PFIM20-10, and PFIM20-30 from NCBI SRA database (accession number: PRJNA794027). The log_10_ of each FPKM value was used to draw a heat map using TBtools (version 1. 0120).

Real-time PCR (qRT-PCR) experiment primer was designed according to CDS sequences of *PfAux/IAA* genes (Appendix A). Total RNA from leaves was extracted by RNAsimple Total RNA Kit (TIANGEN, Beijing, China). Then, the PrimeScript^TM^ RT reagent kit (GenStar, Beijing, China) was used to reverse transcribe RNA into cDNA. The qRT-PCR system consisted of 10 µL 2× RealStar SYBR Mix (Genstar, Beijing, China), 1 µL cDNA template, 1 µL each of upstream and downstream primer, and 8 µL ddH_2_O. The reaction procedure contained 120 s of predenaturation at 95 °C, 15 s of denaturation at 95 °C, 30 s of annealing at 60 °C, and 30 s of extension at 72 °C with 40 cycles. The relative expression values were analyzed using the 2^−ΔΔCt^ method, with actin as the normalized gene. Three biological replicates were performed per sample. Correlations between the qRT-PCR results and transcriptome data were assessed by linear regression analysis using the GraphPad (8.0.2) software.

### 4.7. Protein Interaction Analysis of the PfAux/IAA and PfARF Proteins

Protein interaction predictions were made using STRING database (https://cn.string-db.org/ (accessed on 6 September 2023)) with *Arabidopsis* as the reference plant.

Yeast two-hybrid assay: Primers were designed according to the coding regions of *PfAux/IAA45* and *PfARF13* (Appendix A), the PCR amplified products were connected into the cloning vector (PJET1.2), and the positive clones were sequenced to confirm their gene sequences. The coding sequence of *PfARF13* with the stop codon was cloned and ligated into the pGADT7 vector with the restriction enzyme *EcoR*I and *Bam*HI, and *PfAux/IAA45* was cloned and ligated into the pGBKT7 vector with the restriction enzyme *Nde*I and *BamH*I.

The transformed AH109 yeast cells with pGADT7-*PfARF13* and pGBKT7-*PfAux/IAA45* were cultured on SD/-Trp/-Leu medium at 30 ℃ for 3–5 days, then the samples were cultured on SD/-Trp/-Leu/-Ade/-His (125 ng·mL^−1^) + X-α-gal (40 ng·mL^−1^) medium at 30 ℃ for 3–5 days, and the protein interaction was determined according to the growth status of yeast.

Bimolecular fluorescence complementation (BiFC): Agrobacterium GV3101(pSoup-p19) containing vector ENN-PfARF13 and ECN-PfAux/IAA45 was resuspended with OD600 adjusted to 0.6–0.65. The two bacteria solutions were mixed in a ratio of 1:1 and stored for 2–3 h. Then, the BiFC carrier was injected into the lower epidermis of the leaves of the *N. benthamiana*, and then observed by laser confocal fiberscope 60 h later.

## Figures and Tables

**Figure 1 ijms-25-02260-f001:**
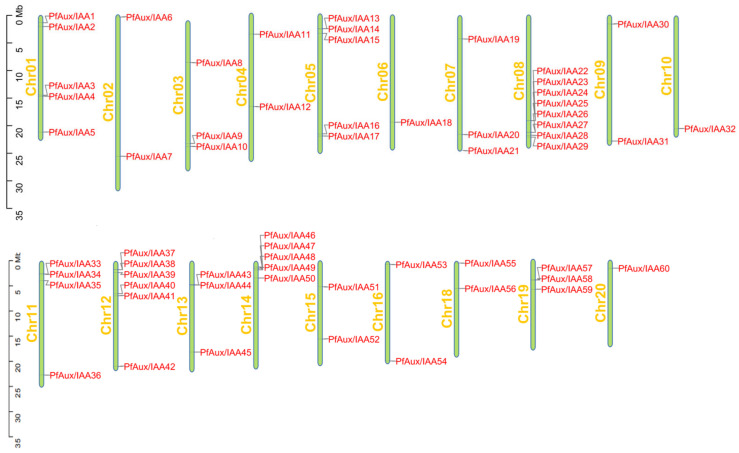
Chromosome localization of *PfAux/IAA* genes in *P. fortunei*.

**Figure 2 ijms-25-02260-f002:**
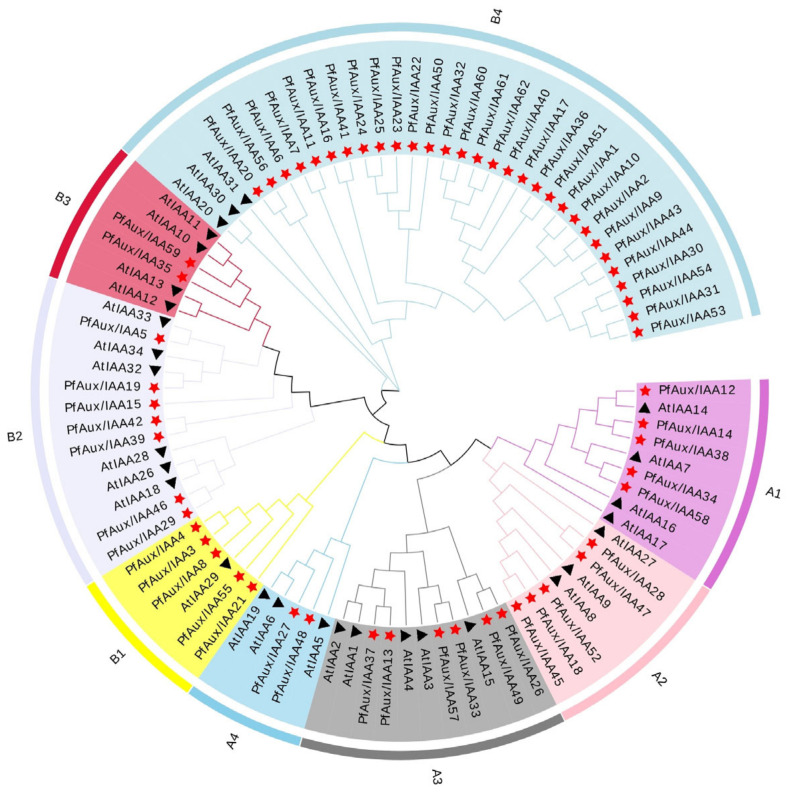
The phylogenetic tree of the *Aux/IAA* genes of *P. fortunei* and *A. thaliana*. The star represents *P. fortunei,* and the triangle represents *A. thaliana.* Subgroups are distinguished by different colors.

**Figure 3 ijms-25-02260-f003:**
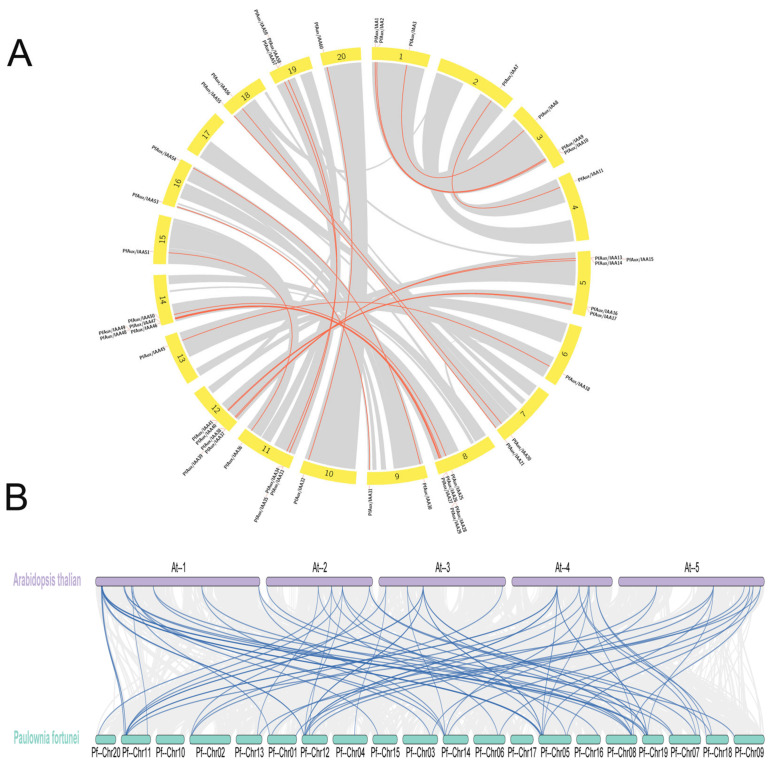
Phylogenetic and synteny analysis of the *Aux/IAA* gene family in *P. fortunei: (***A**) Synteny analysis of *P. fortunei.* Grey lines indicate genomic repeat block. Red lines indicate duplicated *PfAux/IAA* genes. (**B**) Phylogenetic analysis between *P. fortunei* and *A. thaliana*. Grey lines indicate collinear block in *P. fortunei* and *A. thaliana* genomes. Blue lines represent collinear gene pair.

**Figure 4 ijms-25-02260-f004:**
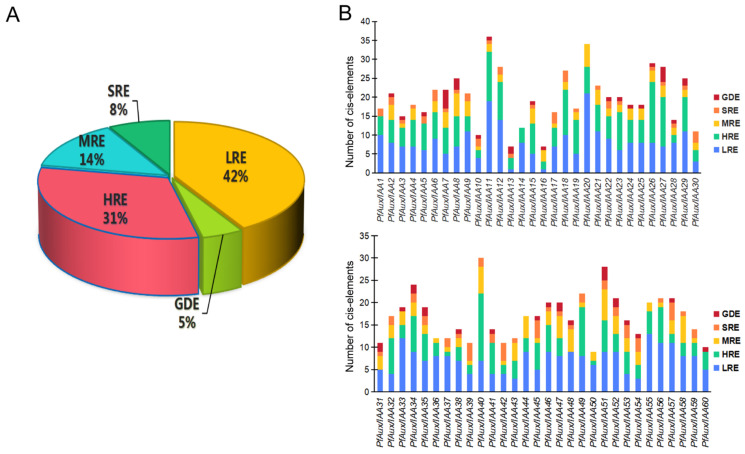
The analysis of cis-elements of the *PfAux/IAA* genes promoter region: (**A**): Different elements were divided into five groups according to biological functions. LRE: light-responsive elements. HRE: hormone-responsive elements. SRE: stress-responsive elements. MRE: metabolic responsive elements. GDE: growth and development elements. (**B**): The cis-elements distribution of individual *PfAux/IAA* genes.

**Figure 5 ijms-25-02260-f005:**
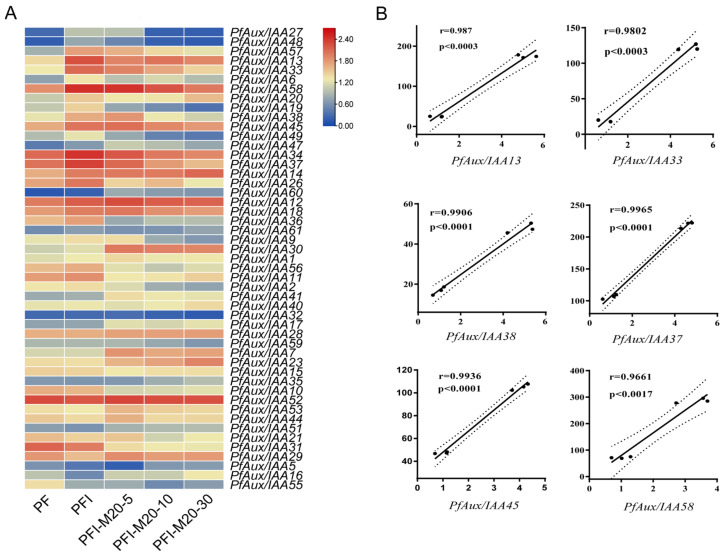
Expression analysis of *PfAux/IAA* genes under phytoplasma infected: (**A**) Heatmap of *PfAux/IAA* genes expression in response to methyl methane sulfonate (MMS) in PaWB diseased seedlings. PF: healthy seedlings. PFI: phytoplasma-infected seedlings. PFI-M20-5: PFI seedlings treated with MMS for 5 days. PFI-M20-10: PFI seedlings treated with MMS for 10 days. PFI-M20-30: PFI seedlings treated with MMS for 30 days. (**B**) Correlation analysis between RNA-seq and qPCR of 6 *PfAux/IAAs*. Pearson correlation analysis was used for statistics; the value of *p* represents significant difference.

**Figure 6 ijms-25-02260-f006:**
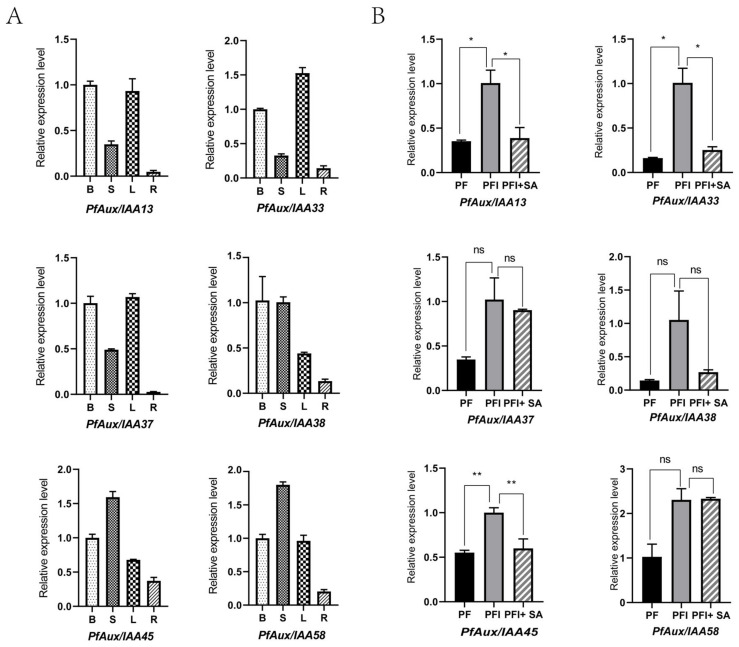
RT-qPCR of *PfAux/IAA* genes expression: (**A**): Expression level of the *PfAux/IAA* genes in different tissues. B: bud; S: stem; L: leaf; R: root. (**B**): Expression level of the *PfAux/IAA* genes under SA treatment. PF: healthy seedlings. PFI: diseased seedlings. PFI+SA: diseased seedlings treated with SA (0.1 mM) for 30 days. Significant differences determined by multiple *t*-tests and indicated by asterisks, * represent *p* < 0.05, ** represent *p* < 0.01, ns present no significant difference. Mean ± showed error bars.

**Figure 7 ijms-25-02260-f007:**
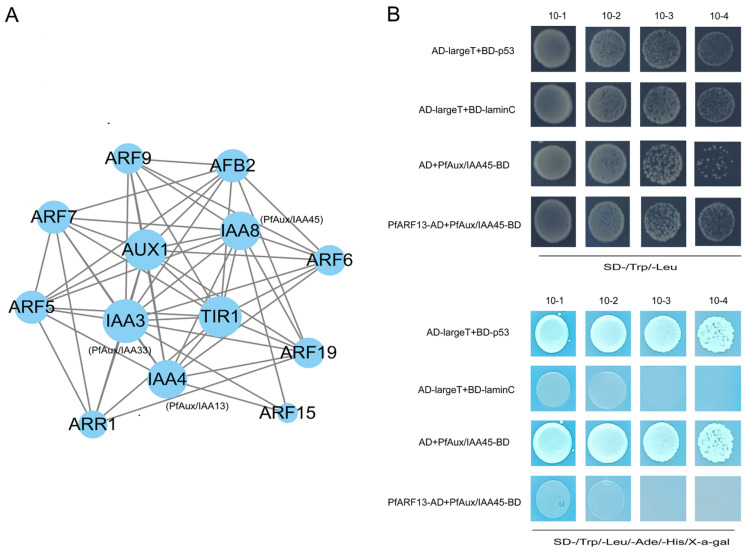
Interaction analysis of PfAux/IAA: (**A**) Protein–protein interaction prediction of PfAux/IAA13/33/45 by STRING database. (**B**) Y2H assay of PfARF13 and PfAux/IAA45. Yeast cells con-transformed with AD-largeT+BD-laminC used as negative control. Yeast cells con-transformed with AD-largeT+BD-p53 used as positive control.

**Figure 8 ijms-25-02260-f008:**
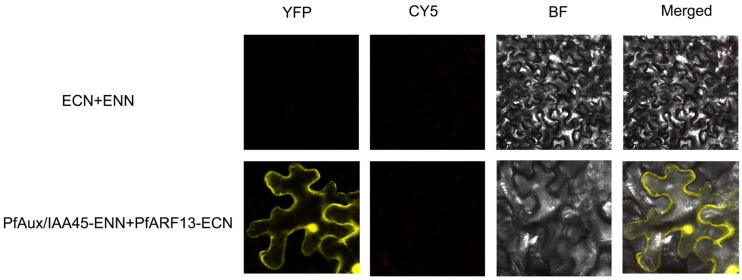
Bimolecular fluorescent complementary of PfARF13 and PfAux/IAA45.

## Data Availability

Data are contained within the article and Appendix A.

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
