# Peer review of "Genome-Wide Identification of the Paulownia fortunei Aux/IAA Gene Family and Its Response to Witches’ Broom Caused by Phytoplasma"

_ijms, 2024, doi:10.3390/ijms25042260_

Round 1

Reviewer 1 Report

Comments and Suggestions for Authors

Article review

1. EVALUATION OF THE PAPER MANUSCRIPT

Title of the manuscript:

"Genome-Scale Identification of the Paulownia fortunei

Aux/IAA Gene Family and its Response to Witches’ broom

cause by Phytoplasma"

Manuscript number: ijms-2803659

The article ijms-2803659 has comprehensively identified and characterized the Aux/IAA gene family in Paulownia fortunei based on various bioinformatics approaches.

Then, expression patterns of the PfAux/IAA genes in seedlings infected by phytoplasmas were explored using qRT-PCR. The article ijms-2803659 is slightly suited to the IJMS as per its aims and scope. However, to improve the quality of the article, some minor points should be noted:

- If possible, the authors should provide a survey of the Aux/IAA family in other higher plant species in the introduction.

- The annotations of the Aux/IAA family, like proteinID and transcriptID are quite lacking. Please improve it.

- The manuscript utilizes genome and proteome data from NCBI, but it does not specify the version of the Paulownia fortunei genome used. For reproducibility and accuracy, it is crucial to include the specific version or release date of the genomic data.

- The prediction of cis-regulatory elements is a very important part. The Results and Discussion section related to this part should concentrate on the stress- and hormone-responsive cis-regulatory elements. Some interesting stress-responsive cis-regulatory elements should be searched in the promoter regions of the PgLAC genes because PlantCARE is an out-of-date tool.

- Why did the authors use the neighbor-joining algorithm to construct the phylogenetic trees? The Maximum Likelihood approach is highly recommended for the generation of phylogenetic trees.

- Authors should clarify the criteria for the protein features (physic-chemical properties) in the method.

- The list of references should exactly follow the journal style.

- Gene name should be italicized. Please, check the whole text.

Comments on the Quality of English Language

- Please, carefully check for grammatical and spelling mistakes. Many typing errors (lack of space) have been easily found in the whole text.

Reviewer 2 Report

Comments and Suggestions for Authors

Comments:

In the present study entitled “Genome-Scale Identification of the Paulownia fortunei Aux/IAA Gene Family and its Response to Witches’ broom cause by Phytoplasma”, authors have comprehensively comprised the systematically identified Aux/IAA gene family members in Paulownia fortunei and their characterization and expression validation in different stress conditions. The study is quite interesting to relate the biological role of Aux/IAA genes particularly in case of phytoplasm pathogen infection in Paulownia fortune plant. However, sometechnical pitfalls are observed which can be rectified to improve the manuscript for global readability and acceptance.  The comments observed during review of the manuscripts are as follows:

Comment #1–Authors should re-write the title of the study. Authors can be re-think about “Genome-Scale” or “Genome-wide”

Comment #2 – The keyword are mostly taken from title itself. In my opinion, the keywords must be largely different from words or phrases mentioned in title so Keyword must be changed and taken from abstract related to major findings and techniques used.    

Comment #3 – Authors should follow uniform style of Aux/IAA. At some places it is not uniformly written. For example, see at line no. 47, 50 and 51

Comment #4 – Authors have stated that “we screened and identified” at line no. 72-73 which make bit confusing statement. What have been screened here is not cleared so it must be clarified or re-write as simple as can be understand.

Comment #5 – The resolutions of some Figure can be improved so that the tags can be seen clearly.

Comment #6 – Regarding ‘Distribution of cis-acting elements in the promoter region of gene family members’ section, authors should quantify the number and distribution of the cis-regulatory elements residing the promoter regions of each gene family members (Fig. 4). For instance, authors can follow and see the pattern of some published manuscript such as Chet Ram, et al. (2022). Genome-wide comprehensive characterization and expression analysis of TLP gene family revealed its responses to hormonal and abiotic stresses in watermelon (Citrullus lanatus). Gene 844 (2022): 146818. Accordingly, the Figure representation can also be re-formulated.

Comment #7 – Authors should rectify PLANCARE or PLANTCARE at line no. 153.

Comment #8 – Authors should write “Planting materials” instead of Plant materials at line no. 286-287.

Comment #9 – It is just suggestion. Authors can incorporate the photographs (if any) taken during treatments in the planting materials either as main Figure or Supplementary data.

Comment #10 – Authors must clarify that whether redundancy in gene family members identified by both methods have been checked? If yes, then it should be mentioned in the methodology section.

Comment #11 – Authors have not cited the references for Tbtools software at line on. 306 and MEGA11 software (line no. 311).

Comment #12 – Authors should rectify the phrase “were taken as default” instead of “were default” at line no. 318.

Comment #13 – It is to rectify that authors have mentioned the 2000 bp up-stream sequences as promoter to analyses the cis-regulatory elements in this study (see line no. 151 and 322). They have used web version of this software. It is noticed that generally the sequence length capacity of this software is only 1500 bp (if used as web tool), then how 2000 bp sequences are analysed. Please clarify the query and same should be rectified in the text of the manuscript. In my opinion, authors should re-analyse the promoter sequences by taking 1500 bp as input sequences.   

Comment #14 – Authors must mention about which normalized gene they have used during qRT-PCR analysis?

Comment #15 – Why authors have not analysed the expression of Aux/IAA gene family members in different tissues of the planting material?

Comment #16 – Authors should describe the PCR protocol with some details like PCR reactions and thermal cycling conditions at line no. 346.

Comment #17 – The discussion section need more elaboration with supportive evidences. It seems to be very concise and few findings such as promoter analysis, gene expression, etc. have not much explained with evidences.

Comment #18 – Authors must expand the conclusion section with quantitative data and touching at least major findings.

Comments on the Quality of English Language

English language is fine however, authors need to re-check the whole contecnt of the manuscript for typological errors.

Reviewer 3 Report

Comments and Suggestions for Authors

Dear Authors,

I had opportunity to review manuscript entitled: “Genome-Scale Identification of the Paulownia fortunei  Aux/IAA Gene Family and its Response to Witches’ broom cause by Phytoplasma” which is considered for publication in IJMS journal. The article introduce new insight in response to Witches’ broom disease and is interesting but need some improvements (mainly in presentation or editoral errors) because amount of needed work in changes I suggest major revision. Comments, I present in a form of list below:

Introduction section:

According IJMS publication rules this section need to precisely formulated aim/ or and hypothesis of the study. Currently no such precisely formulated aim is not present and it must be added.

Result section:

Generally good written but has problems with overloaded Figures which makes in somcases unreadable and too small (too low quality). Problem occurs in case of :

Figure 5 is to small which makes part A and B unreadable. Part B is also too small to see good any results

Figure 6 Below charts with PfAux/IAA38, 45 and 58, I strange line I do not know why

Figure 7 Part a and B must be separated from C. Suggest creation of new Figure only with confocal photos. It makes the figure more clear now part B is unreadable.

Sincerely,

Reviewer 4 Report

Comments and Suggestions for Authors

The manuscript “Genome-Scale Identification of the Paulownia fortunei Aux/IAA Gene Family and its Response to Witches’ broom cause by Phytoplasma” identified auxin-responsive genes Aux/IAA in Paulownia fortunei genome. The Aux/IAA were analyzed by Chromosome localization, phylogenic trees, and cis-acting elements in the promoter regions. Some of the important information was missing from the current paper. Authors need to improve the paper quality.

Major concerns

1.      In line 12, Authors mentioned auxin signaling is related to PaWB phytoplasma symptom. In the previous paper, the symptom is induced by single phytoplasma effector “Tengu”. Authors need to clarify whether PaWB phytoplasma has Tengu in the genome.

2.      Line 158-161 described “It was worth….”. It was not sure why the presence of element responding to jasmonic acid means PfAux/IAA genes’ defense function. Please explain the sentence using previous research.

3.      In line168-186 mentioned methyl methane sulfonate (MMS) treatment. The Figure legends were not correct in Figure 5. In addition, MMS treatment disrupted phytoplasma infection as shown in Authors’ previous paper. That means Authors detected gene expression change bt MMS treatment, not by phytoplasma.

4.      In Figure 6, Authors analyzed PfAux/IAA expression levels under Salicylic acids (SA) treatment. The RNAs were extracted from the infected seedling after 30 days of SA treatment. It is speculated that the long-term SA treatment affects phytoplasma infection. The experiment should have been performed in short-term treatment to analyze the change of PfAux/IAA expression levels by SA.

5.      In Figure 7, Authors investigated PfARF13-PfAux/IAA45 protein-protein interactions. It is not clear why the experiment was needed in the paper.

6.      Line 286-295 about plant material and treatment missed the information about inoculation method and plant condition. Please add a brief description about them, not cite the paper.

Minor concerns

1.      Supplemental data missed Figure legends.

2.      Line 213 and 218 mentioned Figure 8. Please check Figure numbers.

Round 2

Reviewer 3 Report

Comments and Suggestions for Authors

Dear Authors,

All my comments was corrected. I recomdnd publication,

Sincerely,